# Workplace Violence in Healthcare Settings: Work-Related Predictors of Violence Behaviours

**Carla Barros** [1,2,*] **, Rute F. Meneses** [1,2] **, Ana Sani** [1,2,3] **and Pilar Baylina** [4]

1    Faculty of Human and Social Sciences, University Fernando Pessoa (UFP), Praça 9 de Abril, 349,
     4249-004 Porto, Portugal
2    FP-I3ID, Research, Innovation and Development Institute of Fernando Pessoa Foundation,
     Praça 9 de Abril, 349, 4249-004 Porto, Portugal
3    Research Center on Child Studies (CIEC), University of Minho (UM), 4710-057 Braga, Portugal
4    School of Health, Polytechnic Institute of Porto, 4200-072 Porto, Portugal
*    Correspondence: cbarros@ufp.edu.pt; Tel.: +351-914-566-838

**Abstract:** Healthcare workers are exposed to workplace violence such as physical assaults, psychological violence and threats of violence. It is crucial to understand factors associated with workplace violence to prevent and mitigate its consequences. This study aims to identify work-related factors that might influence workplace violence in healthcare settings. A cross-sectional study was developed between March and April of 2022 with healthcare workers. The Aggression and Violence at Work Scale was used to assess workplace violence, and psychosocial risks were assessed through the Health and Work Survey—INSAT. Statistical analysis using bivariate analysis was performed to identify the psychosocial risk factors related to physical violence, psychological violence and vicarious violence. Subsequently, a multiple linear regression was performed to identify the models that better explained the relationship between psychosocial risk factors and the three dimensions of violence. Psychological violence was frequently experienced by the healthcare workers. Significant associations were found between psychosocial risk factors and physical, psychological and vicarious violence, namely working hours, work relationships, employment relations, high demands and work intensity. These findings highlight the importance of taking into consideration work-related factors when designing interventions to prevent and address workplace violence in healthcare settings.

**Keywords:** workplace violence; healthcare settings; risk factors; violence behaviours

## 1. Introduction

Healthcare workers are the most vulnerable professionals to workplace violence [1,2]. Various forms of violence have been reported, such as verbal abuse, threats of violence and physical assault [3–5], typically from patients or patients' relatives, but also between professionals. Violence and harassment affect the health and wellbeing of healthcare workers, namely physical and mental health, compromising work performance and job satisfaction [6–8].

Violence is a complex problem and it is considered a global public health issue that affects the population [9–12]. Workplace violence, particularly in healthcare settings, if not adequately addressed, will become a global phenomenon [13], undermining the harmony and stability of healthcare providers. While verbal violence, physical aggression and intimidation are the most frequent forms of workplace violence recognised, other forms of violence have been increasing, such as humiliation, defamation and offense, in healthcare settings [13–15].

Healthcare workers are exposed to considerable violent behaviours, such as physical violence (consisting of a variety of physically violent behaviours and threats), psychological violence (consisting of several types of psychological aggression at work, stemming from

three sources: colleagues, supervisors and the public) and vicarious violence (consisting of a variety of violent events witnessed or heard about by co-workers, supervisors, friends or relatives), with serious consequences for their health and wellbeing [16–18]. The experience of workplace violence is associated with several adverse health outcomes, including depression, loss of self-esteem, sleep disorders, anxiety, irritability, difficulty concentrating and feeling emotionally upset [19–21]. Furthermore, workplace violence decreases worker motivation and engagement, leading to increased absenteeism, turnover and worker burnout [22,23].

Therefore, it is critical to understand which factors associated with workplace violence can prevent and moderate its consequences, as identified in several studies associated with violence in the work setting [24–26]. The increase in violence in healthcare settings is related to demanding workloads, stress, poor interpersonal relationships and social and economic restraints [14,18,27].

The exposure to different categories of psychosocial risk, including increased workloads, time pressure, communication difficulties and work organisation, high emotional demands, lack of support from staff and management, unsupportive social relationships and ethical and social conflicts at work, impacts healthcare workers' health and wellbeing [28,29]. The pace and intensity of work, interpersonal conflicts and emotional demands can be associated with an increase in aggressive behaviours in the workplace, namely with other professional workers and also with patients [28,30].

In fact, an integrative approach that analyses the interaction between work and organisational variables in the prediction of workplace violence is important to identify work-related factors that can be minimised in order to reduce the impact on individuals and organisations [30,31]. Some studies showed the associating factors that contributed to the enlarged incidence of violence towards healthcare workers over recent years. Risk factors such as long working hours, the pace and intensity of work, emotional demands, poor working relations and interpersonal conflicts and a lack of information and resources can induce violent behaviour [32–35].

Workplace violence is a priority issue and prevention measures must be incremented to ensure physical and mental health and wellbeing. Promoting health and wellbeing amongst healthcare workers is an ethical concern, affecting not only personal health, but also patients and society as a whole, since it affects the quality of healthcare [36,37].

Hence, understanding work-related factors that are associated with workplace violence is central in defining appropriate and targeted interventions to improve workplace safety and health settings. Therefore, the main objective of this study is to analyse the associations between psychosocial risk factors and violent behaviours, namely physical violence, psychological violence and vicarious violence. This study also aims to identify predictors that should be analysed by organisations in order to reduce workplace violence and manage interventions in healthcare settings.

## 2. Materials and Methods

### 2.1. Participants

The sample was composed of 276 healthcare workers—nurses (59.1%), physicians (16.3%), healthcare assistants (13.0%) and administrative assistants (11.6%)—working in hospitals and primary healthcare centres in Portugal. It was composed of 83.3% female and 16.7% male participants, aged between 18 and 71 years ($M = 38.17$; $SD = 10.51$). The majority of the healthcare workers (64.9%) had been practicing for less than 16 years. The majority of the participants (80.1%) worked under permanent contracts.

### 2.2. Measures

#### 2.2.1. Demographics

Participants' demographic data, collected with closed questions, included information about age, gender, marital status, professional activity and years of experience as a healthcare worker.

### 2.2.2. Aggression and Violence at Work

The violence in the workplace was assessed through the Aggression and Violence at Work Scale [30], which evaluates three dimensions of violence: physical violence, psychological violence and vicarious violence at work. The physical violence subscale consists of eight items reflecting a variety of physically violent behaviours and threats (e.g., being hit, kicked, threatened with a weapon). Psychological violence was measured with a three-item subscale representing exposure to psychological aggression at work stemming from three sources: colleagues, supervisors and members of the public (e.g., being yelled at or sworn at). The vicarious violence subscale, with five items, indicates how often they had witnessed or heard about violent events experienced by co-workers, supervisors, friends or relatives. All three dimensions of violence are covered by a total of 16 items arranged in an ordinal Likert-type scale with four classes that indicate a frequency (ranging from 0 for "never" to 3 for "four or more times"). The measure has been shown to have acceptable construct validity and reliability for the Portuguese version used in this study (Cronbach's Alpha > 0.68 for all subscales) [38].

### 2.2.3. Psychosocial Risk Factors

The psychosocial risks were assessed through the Health and Work Survey—INSAT. The INSAT Survey [39] is a self-administered questionnaire that evaluates the relationships between working conditions, risk factors and health problems. Regarding the main goal of this study, only the psychosocial risk factors scale was used. The psychosocial risk factors were the following: work intensity; autonomy; work relationships with co-workers and supervisors; employment relationships with the organisation; emotional demands; ethical and value conflicts. All items are arranged in an ordinal Likert-type scale with five classes that indicate a frequency (ranging from 0 for "not being exposed" to 5 for "being exposed with high discomfort"). The INSAT has good internal consistency, obtained by the Rasch Partial Credit Model analysis, with a Person Separation Reliability coefficient of 0.8761, and it has been used in several health-related studies before [28,40,41].

### 2.3. Procedures

The study involved different scales, starting with a cover sheet with a brief explanation of the purpose of the study, the explanation of the study and the implied consent. Implied consent was obtained by all participants. The estimated time for completing the questionnaire was approximately 10 min. The study was approved by the ethics committee of University Fernando Pessoa (Porto, Portugal, Ref. FCHS/PI-219/21-2), taking into account the procedures of the Declaration of Helsinki. Data were collected online, by sharing a questionnaire through Google Forms between March and April 2022 among 3 hospitals (2 public and 1 private) and 2 primary healthcare centres in Portugal's northern and central regions. Sampling was performed with the snowball technique among healthcare workers from Portugal, with the collaboration of the institutional occupational physician.

### 2.4. Data Analysis

A descriptive statistical analysis on all variables assessed was performed. Frequency and percentage analyses were performed on the demographic characteristics of the participants. Afterwards, all psychosocial risk factors were transformed into nominal variables (0 for "no" answer and 1 for "yes "answer, regarding the level of discomfort) to analyse the associations between risk factors and aggression and violence at work. Then, a bivariate analysis was performed using point-biserial correlation to identify the psychosocial risk factors related to the dependent variables related to violence, particularly physical, psychological and vicarious. Subsequently, a multiple linear regression was used on the statistically significant associations to identify the models that better explained the relationship between psychosocial risk factors and the three dimensions of violence. The regression equations satisfied all assumptions, and the results of the regression analyses

were considered reliable. Data were analysed with the support of the IBM SPSS statistical program for Windows, version 28.0 (SPSS Inc., Chicago, IL, USA).

## 3. Results

The INSAT Psychosocial Risk Factors Scale, presented in Table 1, shows the frequency distribution of "yes" answers to psychosocial risk factors at work that have a significant impact on the healthcare workers' practice. Results show high exposure to psychosocial risks. The pace and intensity of work and emotional demands stand out as risk factors with higher overall mean percentages.

**Table 1.** Descriptive analysis of psychosocial risk factors.

| High Demands and Work Intensity | % Yes |
|---|---|
| Intense work pace | 93.8 |
| Depend on colleagues to do my work | 80.1 |
| Depend on direct clients' requests | 84.8 |
| Have to deal with contradictory instructions | 78.6 |
| Exposed to frequent interruptions | 81.9 |
| Exposed to hyper-solicitation | 83.7 |
| **Working Hours** | **% Yes** |
| Having to go beyond normal working hours | 85.1 |
| Having to sleep at unusual hours because of work | 60.9 |
| Have to "skip" or shorten a meal or not have a break | 83.7 |
| Have to maintain permanent availability | 61.6 |
| **Lack of Autonomy** | **% Yes** |
| Have no freedom to decide how to do my work | 55.1 |
| Restricted schedule without any possibility of change | 42.4 |
| Restricted work break without any choice | 48.9 |
| Not be able to participate in decisions concerning my work | 59.4 |
| **Work Relationships** | **% Yes** |
| Needing help from colleagues and not having it | 43.8 |
| Not having the possibility to exchange experiences with colleagues | 33.3 |
| My opinion being disregarded for the service's functioning | 44.6 |
| Not having recognition by colleagues | 42.0 |
| Not having anyone I can trust | 30.1 |
| Don't feel comfortable in my workspace | 42.8 |
| Impossible to express myself | 39.5 |
| **Employment Relations** | **% Yes** |
| Lack of means to carry out my work | 65.6 |
| In general, I feel exploited | 61.6 |
| There are employment conditions that shake my dignity | 47.5 |
| **Emotional Demands** | **% Yes** |
| Have to deal with situations of tension with the public | 91.3 |
| Being exposed to the suffering of the others | 93.1 |
| Have to simulate good mood and/or empathy | 80.8 |
| Have to hide emotions | 74.6 |
| **Ethical and Value Conflicts** | **% Yes** |
| Have to do things I disapprove | 54.7 |
| My professional conscience is shaken | 46.4 |
| The things I do are considered unimportant | 48.6 |
| Lack the means to do the job well done | 64.1 |

The descriptive analysis for the three dimensions of violence is presented in Table 2.

**Table 2.** Descriptive analysis of physical violence, psychological violence and vicarious violence (*N* = 276).

| Violence Dimensions | *M* (*SD*) | Min.–Max. |
|---|---|---|
| Physical violence | 1.26 (0.98) | 0–3 |
| Psychological violence | 1.54 (1.06) | 0–3 |
| Vicarious violence | 0.52 (0.59) | 0–3 |

The results of the point-biserial analysis are presented in Table 3, with the statistically significant correlations observed between risk factors and violence dimensions.

**Table 3.** Point-biserial analysis: correlations between psychosocial risks and violence dimensions.

| Psychosocial Risk Factors | Violence Dimensions | | | | | |
|---|---|---|---|---|---|---|
| | Physical | | Phychological | | Vicarious | |
| | R | p | R | p | R | p |
| **High Demands and Work Intensity** | | | | | | |
| Intense work pace | 0.166 | 0.006 | 0.270 | <0.001 | 0.251 | <0.001 |
| Depend on colleagues to do my work | 0.194 | 0.001 | 0.197 | <0.001 | 0.149 | 0.013 |
| Have to deal with contradictory instructions | 0.265 | <0.001 | 0.358 | <0.001 | 0.256 | <0.001 |
| Exposed to frequent interruptions | 0.281 | <0.001 | 0.408 | <0.001 | 0.311 | <0.001 |
| Exposed to hyper-solicitation | 0.246 | <0.001 | 0.243 | <0.001 | 0.175 | 0.004 |
| **Working Hours** | | | | | | |
| Having to go beyond normal working hours | 0.109 | 0.071 | 0.129 | 0.033 | 0.156 | 0.009 |
| Having to sleep at unusual hours because of work | 0.338 | <0.001 | 0.357 | <0.001 | 0.372 | <0.001 |
| Have to "skip" or shorten a meal or not have a break | 0.195 | 0.001 | 0.200 | <0.001 | 0.211 | <0.001 |
| Have to maintain permanent availability | 0.246 | <0.001 | 0.184 | 0.002 | 0.187 | 0.002 |
| **Lack of Autonomy** | | | | | | |
| Have no freedom to decide how to do my work | 0.299 | <0.001 | 0.291 | <0.001 | 0.286 | <0.001 |
| Restricted schedule without any possibility of change | 0.312 | <0.001 | 0.213 | <0.001 | 0.217 | <0.001 |
| Restricted work break without any choice | 0.274 | <0.001 | 0.264 | <0.001 | 0.228 | <0.001 |
| Not be able to participate in decisions concerning my work | 0.310 | <0.001 | 0.352 | <0.001 | 0.271 | <0.001 |
| **Work Relationships** | | | | | | |
| Needing help from colleagues and not having it | 0.371 | <0.001 | 0.317 | <0.001 | 0.227 | <0.001 |
| Not having the possibility to exchange experiences with colleagues | 0.302 | <0.001 | 0.187 | 0.002 | 0.112 | 0.042 |
| My opinion being disregarded for the service's functioning | 0.315 | <0.001 | 0.318 | <0.001 | 0.229 | <0.001 |
| Not having recognition by colleagues | 0.184 | 0.002 | 0.153 | 0.011 | 0.105 | 0.044 |
| Not having anyone I can trust | 0.227 | <0.001 | 0.199 | <0.001 | 0.136 | 0.030 |
| Don't feel comfortable in my workspace | 0.280 | <0.001 | 0.258 | <0.001 | 0.161 | 0.008 |
| Impossible to express myself | 0.247 | <0.001 | 0.248 | <0.001 | 0.126 | 0.038 |
| **Employment Relations** | | | | | | |
| Lack of means to carry out my work | 0.273 | <0.001 | 0.381 | <0.001 | 0.346 | <0.001 |
| In general, I feel exploited | 0.273 | <0.001 | 0.381 | <0.001 | 0.346 | <0.001 |
| There are employment conditions that shake my dignity | 0.364 | <0.001 | 0.360 | <0.001 | 0.328 | <0.001 |
| **Emotional Demands** | | | | | | |
| Being exposed to the suffering of the others | 0.134 | 0.035 | 0.289 | <0.001 | 0.187 | 0.002 |
| Have to simulate good mood and/or empathy | 0.241 | <0.001 | 0.344 | <0.001 | 0.270 | <0.001 |
| Have to hide emotions | 0.268 | <0.001 | 0.345 | <0.001 | 0.271 | <0.001 |
| **Ethical and Value Conflicts** | | | | | | |
| Have to do things I disapprove | 0.210 | <0.001 | 0.318 | <0.001 | 0.236 | <0.001 |
| My professional conscience is shaken | 0.278 | <0.001 | 0.364 | <0.001 | 0.306 | <0.001 |
| The things I do are considered unimportant | 0.253 | <0.001 | 0.330 | <0.001 | 0.219 | <0.001 |
| Lack the means to do the job well done | 0.275 | <0.001 | 0.357 | <0.001 | 0.315 | <0.001 |

A multiple linear regression between these psychosocial risk factors and the three violence dimensions was performed to analyse their mathematical relationships, and the statistically significant results that predict physical violence ($F_{(32, 0.95)}$ = 2 960, $p < 0.001$, R = 0.530, $R^2$ = 0.281) psychological violence ($F_{(32, 0.95)}$ = 4.449, $p < 0.001$, R = 0.608,

$R^2 = 0.370$) and vicarious violence (F(32, 0.95) = 3.527, $p < 0.001$, R = 0.563, $R^2 = 0.317$) are presented in Table 4.

**Table 4.** Multiple linear regression analysis between psychosocial risks and violence dimensions.

| Psychosocial Risk Factors vs. Violence Dimensions | Unstandardised Coefficients | | Standardised Coefficients | t Value | *p* | 95.0% Confidence Interval for β | |
|---|---|---|---|---|---|---|---|
| | β | Std. Error | Beta | | | Lower Bound | Upper Bound |
| **Physical** | | | | | | | |
| Having to sleep at unusual hours because of work | 0.189 | 0.087 | 0.155 | 2.182 | 0.030 | 0.018 | 0.360 |
| Needing help from colleagues and not having it | 0.219 | 0.093 | 0.183 | 2.345 | 0.020 | 0.035 | 0.403 |
| There are employment conditions that shake my dignity | 0.171 | 0.096 | 0.144 | 1.785 | 0.046 | −0.018 | 0.359 |
| **Psychological** | | | | | | | |
| Exposed to frequent interruptions | 0.599 | 0.196 | 0.218 | 3.053 | 0.003 | 0.213 | 0.986 |
| Having to sleep at unusual hours because of work | 0.415 | 0.145 | 0.191 | 2.868 | 0.004 | 0.130 | 0.700 |
| Needing help from colleagues and not having it | 0.334 | 0.156 | 0.156 | 2.141 | 0.033 | 0.027 | 0.641 |
| **Vicarious** | | | | | | | |
| Having to sleep at unusual hours because of work | 0.450 | 0.139 | 0.224 | 3.232 | 0.001 | 0.176 | 0.725 |
| Needing help from colleagues and not having it | 0.255 | 0.150 | 0.129 | 1.697 | 0.048 | −0.041 | 0.551 |
| Lack of means to carry out my work | 0.343 | 0.166 | 0.166 | 2.068 | 0.040 | 0.016 | 0.669 |
| There are employment conditions that shake my dignity | 0.318 | 0.154 | 0.162 | 2.067 | 0.040 | 0.015 | 0.621 |

The analysis of the β values and respective *p*-values shows that some cross-sectional psychosocial risks are predictors of all violence dimensions: "Having to sleep at unusual hours because of work" (β = 0.155; *p* = 0.030 for physical violence, β = 0.191; *p* = 0.004 for psychological violence and β = 0.224; *p* = 0.001 for vicarious violence) and "Needing help from colleagues and not having it" (β = 0.183; *p* = 0.020 for physical violence, β = 0.156; *p* = 0.033 for psychological violence and β = 0.129; *p* = 0.048 for vicarious violence).

The psychosocial risk factor "There are employment conditions that shake my dignity" is related to two violence dimensions: physical (β = 0.144; *p* = 0.046) and vicarious (β = 0.162; *p* = 0.040). Finally, there are other psychosocial risk factors related to only one dimension: "Exposed to frequent interruptions", related to psychological violence (β = 0.218; *p* = 0.003), and "Lack of means to carry out my work", related to vicarious violence (β = 0.166; *p* = 0.040).

The positive β values corresponding to the significant predictors allow us to conclude that exposure to psychosocial risk factors is related to violence dimensions. The strength of the different predictors in this model is very similar since the β values are identical.

## 4. Discussion

Workplace violence is a complex and serious occupational issue leading to adverse mental and physical health outcomes in all involved. Furthermore, work-related and behavioural factors may be associated with witnessing and/or experiencing workplace violence among healthcare workers. Therefore, this is a line of research with significant social implications.

The results obtained in the present study revealed high exposure to psychosocial risks, in accordance to previous studies [14,18,27–29], namely working hours, work relationships, employment relations and high demands and work intensity. In line with former

research [3–5,13–15], this study also showed that violence, mostly psychological violence (among the three types assessed), was frequently experienced by the healthcare workers of the sample.

Moreover, statistically significant associations were found between psychosocial risk factors and physical, psychological and vicarious violence, strengthening preceding findings [13,18,27].

It was thus possible to identify, among the psychosocial risks considered, significant predictors of the different types of violence assessed. These results suggest some predictors that should be taken carefully into consideration by healthcare organisations in order to reduce workplace violence (and its complex and vast consequences): working hours (and its impact on sleep), work relationships (namely, colleagues' help), employment relations (specifically, employment conditions and lack of means) and frequent interruptions. These are in accordance with previous studies [24,26,42].

These results also suggest that the key psychosocial risk factors identified in this study might contribute to a lower ability for effective conflict management among healthcare workers. Thus, better knowledge of working conditions would allow the development of occupational health interventions and better organisational support, leading to a reduction in workplace aggression and violence [42–45].

Consequently, it is crucial to develop and optimise interventions focusing on psychosocial risk factors as a way to decrease violence in healthcare settings. The differential efficacy of these interventions in specific groups of healthcare workers should also be tested in order to identify what works best for whom. New studies must be carried out on how the different healthcare groups cope with workplace violence, taking into consideration other variables such as sex and age differences.

## 5. Conclusions

The findings of this research add value to the research field by highlighting the importance of work-related risk factors and their role in experiencing workplace violence. Namely, high exposure to psychosocial risks, such as the pace and intensity of work, work relationships and emotional demands, have a high impact on violent behaviours.

Workplace violence needs a holistic approach. The analysis of psychosocial risk factors should be the basis for developing occupational prevention and intervention strategies among healthcare providers. Awareness and recognition, followed by work environment prevention measures, are essential, considering the comprehensive impact on patient safety. These measures are also conditional to ensure that healthcare workers are in a safe working environment, promoting better work satisfaction and better and safer healthcare services for the population. However, it is acknowledged that the study has some limitations that may have influenced some of the results, such as the sample size, which limited the performance of a gender analysis, age difference evaluation or comparison between professional groups. These variables should be explored due to social, organisational and culture issues. Therefore, a larger sample must be considered in future research on workplace violence.

**Author Contributions:** Conceptualisation, C.B., A.S. and R.F.M.; Methodology, C.B., A.S. and R.F.M.; Formal analysis, P.B.; Investigation, C.B., A.S. and R.F.M.; Writing—original draft preparation, C.B., P.B., A.S. and R.F.M.; Writing—review and editing, C.B., A.S. and R.F.M. All authors have read and agreed to the published version of the manuscript.

**Funding:** This research received no external funding.

**Institutional Review Board Statement:** The study was conducted according to the guidelines of the Declaration of Helsinki and approved by the Institutional Ethics Committee of Fernando Pessoa University (protocol code Ref. FCHS/PI 219/21-2 and date of approval 19 January 2022).

**Informed Consent Statement:** Informed consent was obtained from all subjects involved in the study.

**Data Availability Statement:** The data presented in this article are not readily available since they were not approved to be shared outside of the research team. Requests to access the datasets should be directed to cbarros@ufp.edu.pt.

**Conflicts of Interest:** The authors declare no conflict of interest.

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
