# Peer review of "Workplace Violence in Healthcare Settings: Work-Related Predictors of Violence Behaviours"

_psych, doi:10.3390/psych4030039_

Round 1

Reviewer 1 Report

You have to improve the explantions about Healthcare workers accoding to gender differencies. You avoid gender variable. It is very relevant because as you say 83,3 % of the participantas are female. You need to explain also why the mayority of people are women and if there are equality in the salaries if we compare with men. This is a key field in this kind of job. The concept of "risk population" needs better explanation. We need to know also which is the criterium to select the participants. Conclusios should be improved according to gender issues and ways of prevention.

Author Response

Dear Reviewer,

Please find attached the reply to the review report,

Kind regards,

Carla Barros

Reviewer 2 Report

The article is clear, succinct, and well-written. The literature review is strong. A sample size of 276 healthcare workers is good.

The major concern about this article is the willingness of the authors to lump 4 distinct jobs into one category: nurses, physicians, healthcare assistants, and administrative assistants. These are very, very different jobs that offer very different exposure to patients, their families, and thus to workplace violence. Physicians in hospitals may have less exposure than the other 3 groups, although it is likely to be the most intense relationship for patients and families. Physicians may or may not have to work required hours. For example, emergency department physicians may work shifts but primary care doctors may just visit patients in hospitals or hold office hours and have far more control over their work schedules. Nurses have considerable exposure because they work intimately with patients over required shifts. Health care assistants and administrative assistants are typically required to work shifts as well.

The authors themselves mention the need to study different healthcare groups. It looks as though they possess the data to begin the work of distinguishing between the experiences of the 4 health care groups. Right now the article basically states work is stressful and difficult for all healthcare workers. The authors can make a stronger contribution by making further analyses according to job title.

The conclusion of the article is relatively weak. It can be strengthened by more specific recommendations for other researchers.

Author Response

(The authors gave the same response as above.)

Round 2

Reviewer 1 Report

You should explain as a limitation the lack of gender diferences.

Reviewer 2 Report

the authors responded to the major issue of not analyzing data by job title.
